# Estimation of annual runoff using supraglacial channel geometry derived from UAV surveys of Qiyi Glacier, northern Tibetan Plateau

Longjiang Xie[1,2], Yuwei Wu[1,2], Ninglian Wang[1,2], Anan Chen[1,2,3], Shiqiang Zhang[1], and Sheng Hu[1]

[1]Shaanxi Key Laboratory of Earth Surface System and Environmental Carrying Capacity, College of Urban and Environmental Science, Northwest University, Xi'an 710127, China

[2]Qiyi Glacier Station, College of Urban and Environmental Science, Northwest University, Xi'an 710127, China

[3]National Cryosphere Desert Data Center, Northwest Institute of Eco-Environment and Resources, Chinese Academy of Sciences, Lanzhou 730000, China.

*Correspondence to*: Yuwei Wu (yuwei.wu@nwu.edu.cn), Sheng Hu (husheng@nwu.edu.cn)

**Abstract.** Due to difficulties in direct field observation and uncertainties in glacier runoff models, accurately estimating the glacier runoff remains one of the foremost challenges in cryospheric science. Using a digital elevation model (DEM) and orthophotos (both with a resolution of 5 cm) obtained from an unmanned aerial vehicle (UAV), this study developed a novel remote sensing method for estimating the annual discharge of the supraglacial channel over Qiyi Glacier in the northern Tibetan Plateau, which contributes to the majority of the glacier runoff. Our results showed that the catchment areas of the six main supraglacial channels covered 92.02% of the total glacier area and transported 89.43% of the annual surface meltwater yield (each ranging from $0.07 \times 10^6$ to $0.66 \times 10^6$ m$^3$). Some geometric parameters of the supraglacial channels (including lateral deviation, gradient, and width) were selected to predict the annual discharge using a stepwise regression model, which explained ~78.2% of the variance in the measurement-based glacier annual discharge, with the explained variance increasing to 81.8% after five-point moving average filtering. In comparison, a nonlinear regression model incorporating only the lateral deviation and specific gradient, which were more easily obtained practically, performed somewhat less well, accounting for 66.2% of the discharge variation; however, the explained variance increased to 81.4% after five-point filtering. If satellite remote sensing data with meter-level spatial resolution are available for a specific glacier research area, our regression models, based solely on the UAV-derived supraglacial channel network, will be a promising solution for monitoring changes in annual glacier discharge.

## 1 Introduction

In the context of global warming, most mountain glaciers worldwide are suffering from accelerated retreat with substantial influences on the glacier-fed rivers, especially for the "water tower of Asia" (Tibetan Plateau), which seriously threatens the livelihoods of hundreds of millions of people downstream with rising risks to water resource safety (Bolch et al., 2012; Huss, 2011; Immerzeel et al., 2020; Nie et al., 2021; Wang et al., 2021; Yao et al., 2022; Zemp et al., 2019). Emergent remote sensing data from different types of spaceborne sensors have been widely used to study various aspects of the influence of climate change on glaciers (Paul et al., 2015; Xu et al., 2025), such as glacier area (Burns and Nolin, 2014), ice mass change (Curran

et al., 2025; Kääb et al., 2012), equilibrium line altitude (ELA) (Rabatel et al., 2013), glacier surface velocity (Dehecq et al.,
2019; Kraaijenbrink et al., 2016), and glacier surface temperature (Ren et al., 2024; Wu et al., 2019). However, there is still a
lack of remote sensing retrieval methods for glacier runoff that link glacier changes directly to the water supply for downstream
human communities, which has become a major obstacle for understanding the patterns of glacier runoff changes for practical
water resource management and policy-making at the watershed or larger scale.
The supraglacial stream and channel network, one of the most common landscapes on the surfaces of continental ice sheets
and mountain glaciers, is formed by the incision and erosion effects of glacial meltwater, which has been shown to have river
morphology and processes similar to those of terrestrially based bedrock rivers (Knighton, 1981; Pitcher and Smith, 2019)
with the discharge closely related to the channel geometry (e.g., width and depth) (Jarosch and Gudmundsson, 2012). However,
compared with the relatively sophisticated methods for estimating terrestrial river runoff by retrieving the water level, river
width, channel slope, and flow velocity from satellite remote sensing data (Bjerklie et al., 2018; Xu et al., 2024), the retrieval
methods for meltwater runoff of mountain glaciers remain less developed, mainly for two reasons. First, most supraglacial
rivers of mountain glaciers are very narrow (sub-meter or meter scale), making it difficult to be identified from the commonly
used satellite images with meters or tens of meters pixel sizes, even though these satellite images have been widely used to
extract the spatial distribution of relatively wider supraglacial rivers on the Greenland Ice Sheet (GrIS) and other large polar
glaciers (e.g., Bell et al., 2017; Legleiter et al., 2014; Yang and Smith, 2016; Zhang et al., 2023). Second, it remains challenging
to obtain continuous field measurements or estimations of supraglacial river discharge from those ice sheets or mountain
glaciers under relatively harsh environmental and climatic conditions (Gleason et al., 2016). To date, estimates of meltwater
runoff at basin or larger scales have often relied either on the limited field observations (Gleason et al., 2016; Smith et al.,
2017, 2021) or on glacier runoff models driven by coarse-resolution climate data (Beamer et al., 2016; Hock, 2005; Sicart et
al., 2008; Wang et al., 2024; Yang et al., 2025). Although in situ meteorological observations from weather stations are used
in some glacier studies, such data are often unavailable or sparse for remote glacierized regions, leading to uncertainties in
modeled runoff that hinder a robust quantitative analysis of its relationship with supraglacial channel geometry (Smith et al.,
2017). Furthermore, during the ablation season, supraglacial channels can terminate in meltwater lakes, moulins, and crevasses
(Colgan et al., 2011; Smith et al., 2015; Yang and Smith, 2016), further complicating the estimation of meltwater runoff of
supraglacial rivers.
Currently, uncrewed aerial vehicle (UAV) remote sensing has been widely used to monitor glacier changes by combining its
optical images and digital elevation model (DEM) data or airborne synthetic aperture radar (SAR) images, which are performed
at a centimeter-level spatial resolution (Bhardwaj et al., 2016; Dall'Asta et al., 2017; Hugenholtz et al., 2013; Immerzeel et al.,
2014; Kraaijenbrink and Immerzeel, 2025; Śledź et al., 2021; Wang et al., 2025). Although UAVs are generally equipped with
visible light cameras, which have limited ability to discriminate water bodies from the surrounding narrow and deep
supraglacial channels on mountain glaciers, UAV remote sensing data have great strengths in identifying supraglacial channel
networks and their cross-sectional geometry. According to studies based on UAV field observations, channel migration and
curvature adjustments occur annually in supraglacial rivers in polar glaciers (Bergstrom et al., 2021; Rippin et al., 2015; St
Germain and Moorman, 2019). Similarly, significant interannual variability in supraglacial rivers has also been reported in the
Ganglongjiama Glacier on the central Tibetan Plateau using a terrestrial laser scanner (TLS) data (Xue et al., 2024). Our field
observations at the Qiyi Glacier also revealed that supraglacial channel adjustment appears to be positively related to the
intensity of ice melting on an annual timescale: years of more intense melting are generally characterized by stronger surface
meltwater erosion with deeper and wider channels, and vice versa. This phenomenon inspired us to investigate the relationship
between the channel morphology and annual discharge of supraglacial rivers. Therefore, in this study, using a DJI M300 series
UAV, a 5 cm spatial resolution DEM and orthomosaic map over the Qiyi Glacier were obtained to extract the channel
geometric parameters at different elevations, which were used to explore their relationships with meltwater discharge
calculated by field observations rather than the glacier runoff model.
**2 Data and methods**
**2.1 Study area**
The Qiyi Glacier, located in the Qilian Mountains (39°14′13″N, 97°45′18″E), is a typical polythermal mountain glacier of the
Tibetan Plateau (Fig. 1). The elevation of the Qiyi Glacier ranges from 4,322 to 5,158 m.a.s.l., with a total area of approximately
2.69 km$^2$ and length of 2.9 km in 2023. Since 1975, the glacier area has shrunk by approximately 4.5%, with the terminus
continuously receding by 235 m. Mountain glaciers (including the Qiyi Glacier) offer important water resources to the
downstream arid regions over the Hexi Corridor, with glacier meltwater accounting for approximately 20% of the runoff of
the Beida River (Wang et al., 2017). We have undertaken regular meteorological, hydrological, and mass balance observations
on Qiyi Glacier from June to August since 2002, which have demonstrated an increase in melt, particularly since 2016 (Chen
et al., 2024). Over the past decade, the acceleration of glacial melting has increased surface meltwater runoff, which has
contributed to the deepening and widening of supraglacial stream channels.

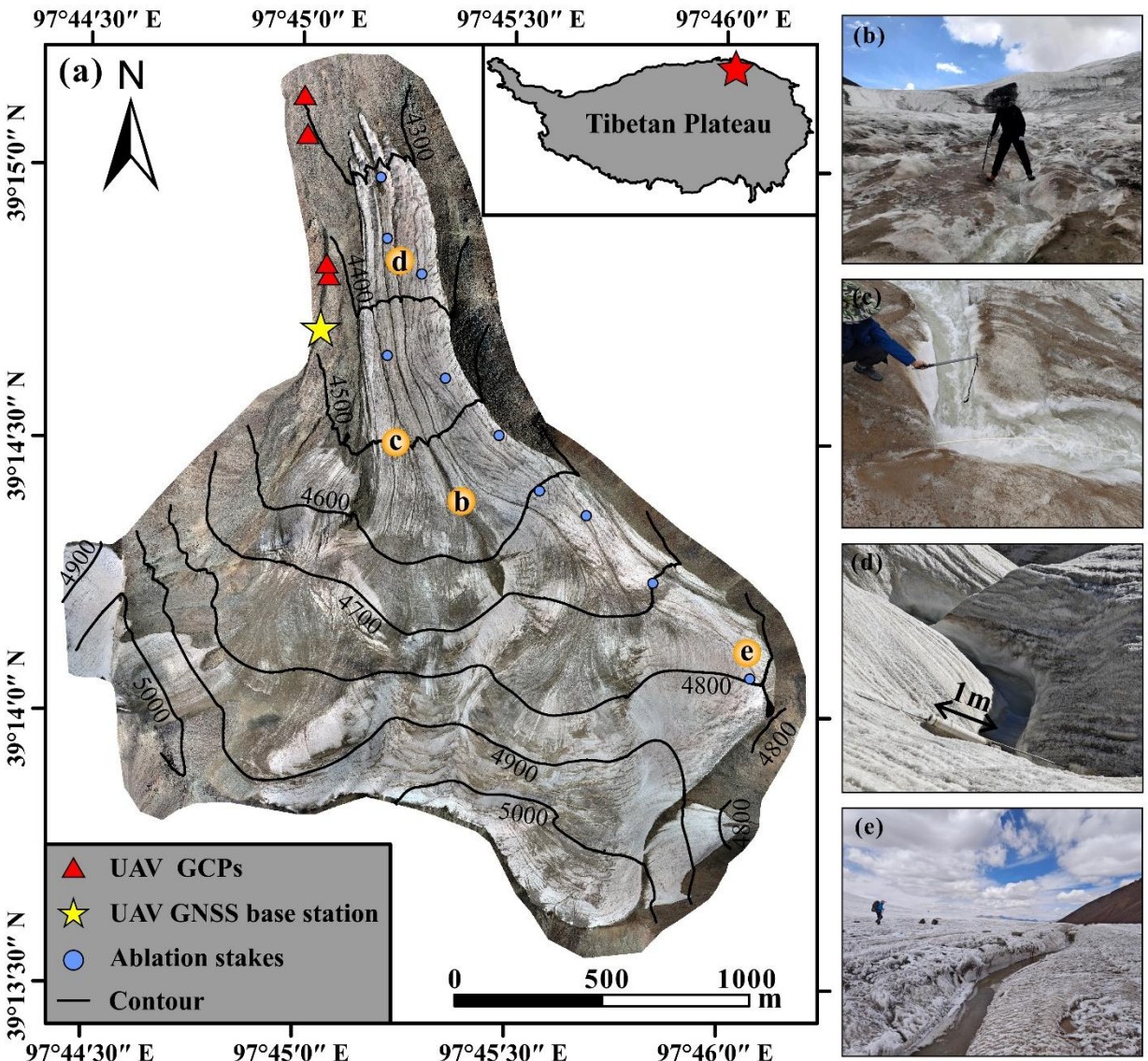

**Figure 1.** Location of the Qiyi Glacier and photos of its typical supraglacial stream channels. (a) Orthomosaic map of the Qiyi Glacier generated from UAV data (5 cm spatial resolution), with red triangles marking the locations of ground control points (GCPs) each with a fixed solution for UAV flights, yellow stars indicating the Global Navigation Satellite System (GNSS) base station of UAV, blue dots representing the locations of ablation stakes on the glacier, and orange circles showing the locations where photos (b)–(e) were taken.

## 2.2 UAV data acquisition and processing

A DJI Matrice300 real time kinematic (RTK) UAV was used to produce the DEM and orthophotos (both with a resolution of 5 cm) of the Qiyi Glacier. The UAV was equipped with a Zenmuse L1 lens integrating a visible-light camera and a light detection and ranging (LiDAR) sensor. From August 15 to August 20, 2023, 2,585 photographs and LiDAR data were collected

at a flight altitude of 100 m to cover the entire Qiyi Glacier. The geolocation accuracy requirements of the DEM and
orthomosaic map were ensured by ground control points (GCPs), and Hi-Target RTK was used to obtain fixed solutions for
high-precision geolocation information.
Pix4D Mapper 4.5.6 software was used to process the visible light images, including data preprocessing, aerial triangulation,
and point cloud encryption. The root mean square errors (RMSE) in $x$, $y$, and $z$ directions for the positions of the GCPs were
0.24m, 0.09m, and 0.35m, respectively. The LiDAR data were processed using the CloudCompare and Lidar360 software,
including denoising, smoothing, thinning, and rasterizing the point cloud. It should be noted it is relatively difficult to
identify homologous image points on the snow surface of the upper glacier under visible light, and the density of the point
cloud data generated from visible light images is very low, while for the photons actively emitted by the lidar sensor, due to
their property of being easily absorbed by the liquid water, the density of the point cloud is very low in the region of the lower
glacier where the melting is strong. Therefore, these two types of point-cloud data have good spatial complementarity for
generating a complete high-precision point-cloud dataset for glacier surfaces. The iterative closest point (ICP) algorithm was
applied to register these two, and the merged point-cloud data were rasterized to the final DEM with a spatial resolution of
0.05 m.

## 2.3 Extraction of morphological characteristics of supraglacial channels

The supraglacial channel network was extracted from UAV-derived DEM data by using the hydrological analysis module of
the ArcGIS platform and Global Mapper software, from which we extracted several morphological characteristics, including
channel sinuosity (*SIN*), lateral deviation (*LD*), channel gradient (*G*), and cross-section geometry (width *W*, depth $\Delta h$, and
width/depth ratio) (Fig. 2). To ensure consistency in defining channel geometry, we applied a standardized approach to
determine channel width and depth. Since supraglacial channels are continuously eroded by rapid water flow, there are
relatively distinct inflection points in the slope of the cross-section. The two points on the cross-section with the steepest slope
gradients were designated as $h_1$ and $h_2$. Channel depth ($\Delta h$) was defined as the average vertical distance from these two points
to the lowest point of the channel, whereas channel width (*W*) was defined as the horizontal distance between $h_1$ and $h_2$. A
representative cross-section (channel A at 4350 m) has been included in the Supplement (Fig. S1) to illustrate the positions of
$h_1$ and $h_2$.
Supraglacial rivers can terminate in crevasses and moulins, potentially leading to the misidentification of channels. Although
crevasses also exist in the Qiyi Glacier, they are generally small in size and mainly distributed in the steep area of the upper
glacier as well as the edge area of the middle and lower parts; therefore, they have a minor influence on the process of river
confluence and continuity of the river channel (Fig. 1). After several comparative trials, a minimum drainage area of 0.02 km$^2$
was determined as the threshold for the automatic extraction method. The automated algorithm identified a total of 33 channel
segments, including not only the six main channels (labeled A–F in Fig. 4a) but also smaller, low-order tributaries. To ensure
the realism of the extracted channels, all automatically derived streams were carefully compared with the orthophotos, channels
that could not be clearly identified in the orthophotos were truncated, particularly small tributaries and river sections at the
upstream headwaters. Finally, six main supraglacial channels on the Qiyi Glacier were correctly identified and extracted, from
which the glacier was divided into six catchments (Fig. 4a).

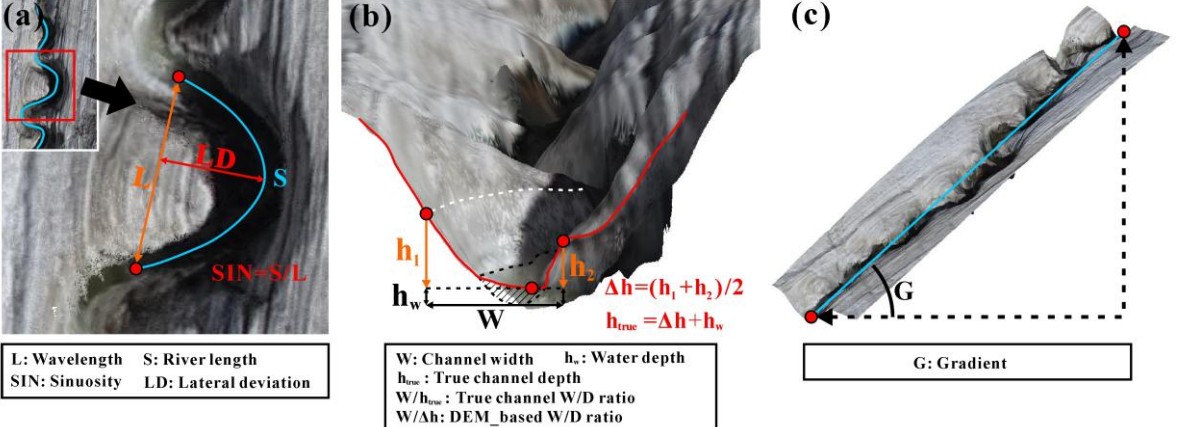


**Figure 2.** Diagrams showing the process of obtaining the channel geometric parameters: (a) sinuosity and lateral deviation of an individual
meander; (b) geometry of the channel profile (width, depth, and width/depth ratio); the white dashed line indicates the maximum slope, and
the black dashed line and shaded area indicate the water surface and underwater channel profile, respectively; (c) the channel gradient.
**2.4 Calculation of the annual glacial discharge transported by supraglacial channels**
Excluding the negligible glacier mass loss due to surface sublimation, the annual meltwater yield of a given pixel (5 cm × 5
cm) on the glacier can be estimated as follows:
$Q_a = (P_0 - MB + (H - 4310) \times 6.8/100) \times S,$                                                      (1)
where $P_0$ is the precipitation at the glacier terminus based on field observations, *MB* and *H* represent the glacier mass balance
and the elevation of the pixel, mass balance is expressed in millimeters water equivalent (mm w.e.), with negative values
indicating mass loss. 4,310 (m) means the elevation of the glacier terminus, *S* denotes the area of the pixel, and 6.8 mm/(100m)
is the precipitation gradient regressed from rain gauge networks on the Qiyi Glacier (Jiang et al., 2010).
From the nine ablation stakes across the glacier with long-term mass balance (*MB*) measurements, the *MB* of all pixels was
calculated using spatial extrapolation (Fig. 3a). The stakes were measured at annual intervals. Several new ablation stakes were
installed on July 20, 2022, and subsequently measured and maintained on August 24, 2023. Detailed information for each
stake data, including stake ID, position, elevation, and mass balance records, are provided in Supplement (Table S1). The
uncertainty of the mass balance based on the ablation stakes is discussed in detail in Section 4.2.
Because our rain gauge observations at the glacier terminus were not collected every year, precipitation data from a neighboring
weather station (Tuole Station, ~70 km from the Qiyi Glacier) were used to establish a relationship to reconstruct the
precipitation at the glacier terminus. Comparisons between two sets of precipitation data during the period 2006–2013 (Fig.
3b) showed a good linear relationship, with the coefficient of determination ($R^2$) value as high as 0.967; therefore, the annual
precipitation ($P_0$) of 2023 at the glacier terminus (4,310 m) can be inferred from the precipitation data at the Tuole station ($P_t$)
as follows:
$P_0 = 1.1267 \times P_t + 15.205.$         (2)
Equations 1 and 2 were used to calculate the meltwater yield for each pixel. From the upper to the lower glacier, the volume
of discharge transported by the ice channel at each 10 m elevation decrease can be determined from watershed analysis by
placing outlets at each elevation interval (10 m).

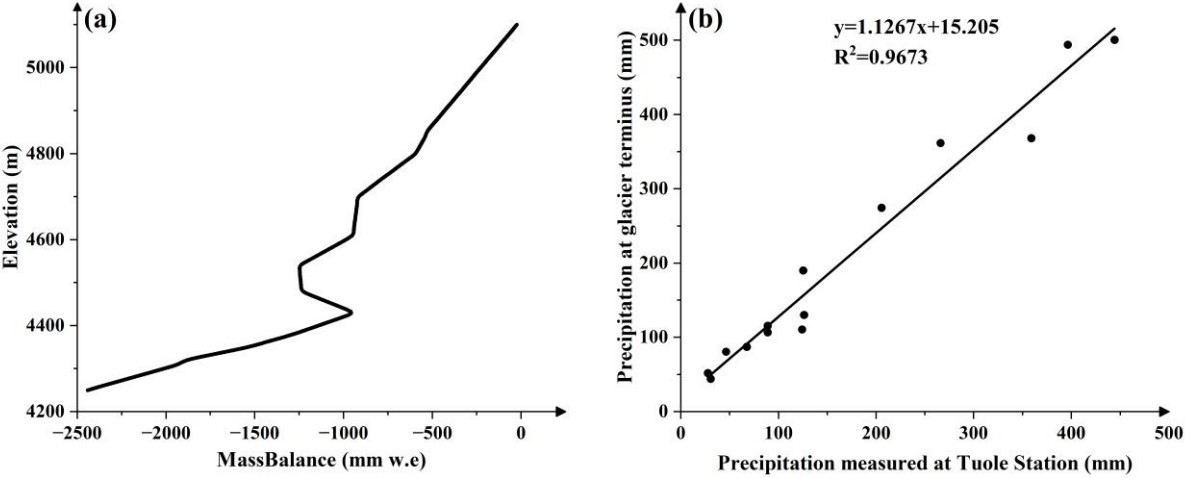


**Figure 3.** (a) Variations in mass balance of the Qiyi Glacier reconstructed from ablation stake measurements with elevation in 2023. (b)
Comparisons of monthly/annual precipitation between Tuole station records and rain gauge observations at the terminus of Qiyi Glacier
during 2006–2013.
**2.5 Statistical modeling methods**
Pearson's correlation ($r$) and Spearman's rank correlation ($\rho$) were used to analyze the correlation between the morphological
characteristics of the supraglacial channels and the annual meltwater discharge transported by these channels at different
elevations. Because previous studies found that the total energy of the flowing water (including pressure energy, gravitational
potential energy, and kinetic energy) is the initial driving force for the channel formation (Rodríguez–Iturbe et al., 1992) and
that discharge and slope are the most influential fluvial factors (St Germain and Moorman, 2019), here we took the product of
gradient $G$ and annual discharge $Q_a$ as the water potential energy factor $GQ_a$ in the correlation analysis.
In addition to the water flow conditions (discharge and water potential energy), the morphology of supraglacial channels is
influenced by other factors, such as pre-existing ice structure (e.g. fractures and moulins), thermal differences (ice and water
temperatures and channel frictional heat), topographic relief (slope, roughness, and surface albedo), and ice velocity (Pitcher
and Smith, 2019). To eliminate the influence of discharge and water potential energy, $Q_a$ and $GQ_a$ were used as control
variables to conduct a partial correlation analysis for each pair of morphological characteristics.
The fitting model was developed using only morphological characteristics that showed a strong correlation with the discharge,
and the regression methods included simple linear regression, stepwise regression, and nonlinear least squares (NLS). The
coefficient of determination ($R^2$), root mean square error (RMSE), mean absolute error (MAE), and normalized root mean
square error (nRMSE) evaluated the performance of the fitting models. Specific details of the regression model and error
analysis can be found in Sections 3.4 and 4.2.

## 3 Results

### 3.1 Supraglacial channel network and annual discharge

The watersheds of the six main supraglacial channels covered 92.02% of the total area of the Qiyi Glacier (Fig. 4a) with an
annual discharge volume of about $2.4 \times 10^6$ m$^3$ (each ranging from $0.07–0.66 \times 10^6$ m$^3$), which contributed 89.43% to the total
meltwater of the glacier surface ($2.68 \times 10^6$ m$^3$) during the hydrological year from August 2022 to August 2023. Approximately
63.21% of the channel discharge was transported to the glacial terminus (by channels A, B, C, and D). In particular, the
discharge in the lower reaches of channels A and C was significantly higher than that in the other channels (Fig. 4b), accounting
for approximately 45.30% of the total glacial discharge. Channel F was the only river that was newly formed during the
summer of 2023 because two separate smaller catchments merged into one larger catchment, reflecting how supraglacial rivers
reshape the surface topography in a short period as meltwater flow increases.

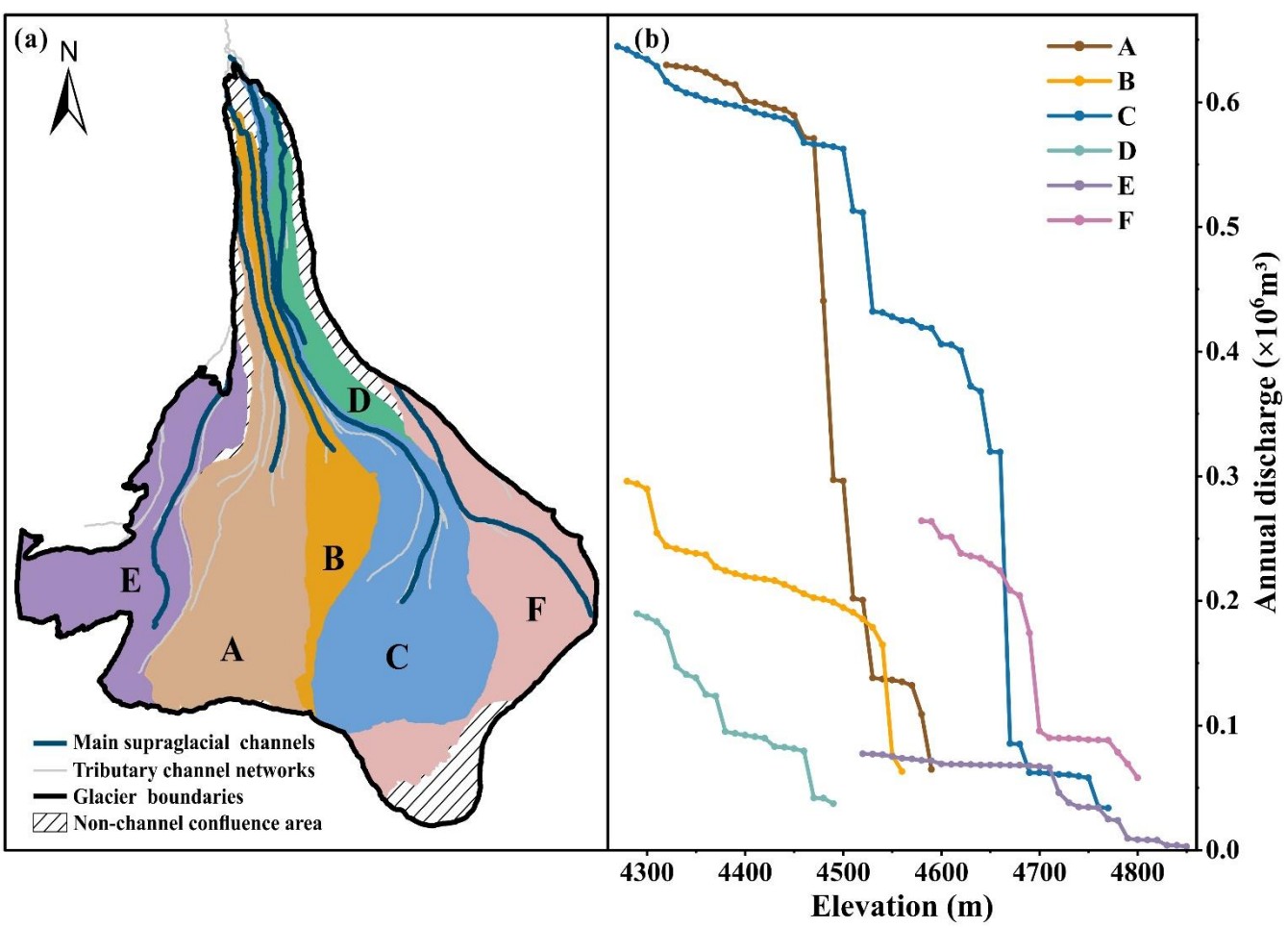

**Figure 4.** (a) Supraglacial channel network over the Qiyi Glacier and (b) variations in the annual discharge of six main supraglacial channels with elevation changes at a 10 m elevation interval. The catchment areas of the main channels are filled with different colors.

## 3.2 Spatial variability of morphological characteristics of supraglacial channels

### 3.2.1 Variations in the sinuosity and lateral deviation

The mean sinuosity and mean lateral deviation integrated with a 10 m elevation interval show similar variation trends among the six main supraglacial channels, and both are lower in the upper reaches and higher in the lower reaches (Fig. 5a and 5b). The median of 10 m average sinuosity and lateral deviation ranges from 1.05 to 1.27 and 0.42 to 1.06, respectively. The maximum sinuosity and lateral deviation were observed near the outfalls of channels C (2.75) and D (4.41), respectively. In general, channels with relatively higher discharge (channels A and C) tended to have larger sinuosity and lateral deviation in both their median and mean, while those with relatively lower discharge (e.g., channels D and E) had lower sinuosity and

lateral deviation (Fig. 5c), suggesting a possible positive correlation between the discharge and sinuosity/lateral deviation of
the supraglacial channel.

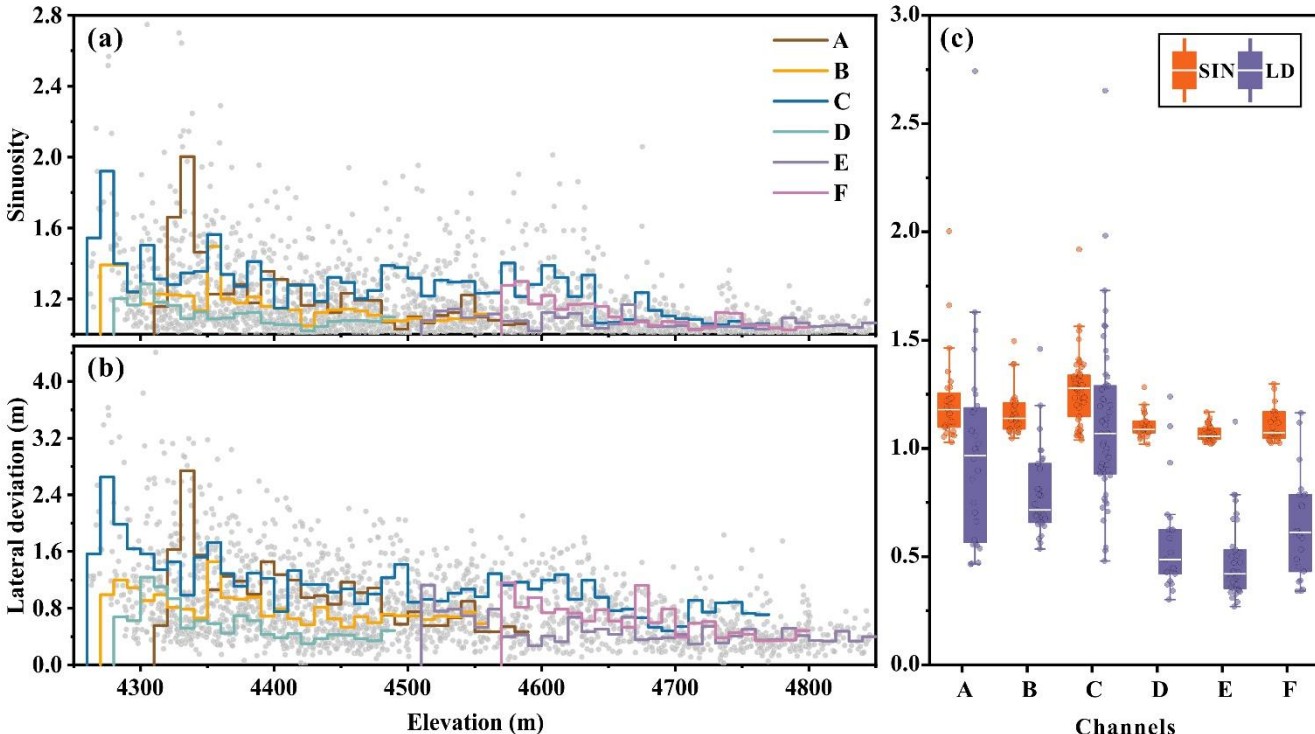


**Figure 5.** Variations in the (a) sinuosity and (b) lateral deviation of six supraglacial channels (gray dots indicate values at each meandering)
and their mean at a 10 m elevation interval (colored stairs); (c) box plot of 10 m average sinuosity and lateral deviation for six supraglacial
channels, where the box represents the interquartile range (IQR), the whiskers denote 1.5 times the IQR from both ends of the box, and the
white horizontal line shows the median; the labeling A to F corresponds to channels A to F as shown in Fig.4a.

### 3.2.2 Variations in the channel gradient


The incision depth (no more than 1 m) of supraglacial rivers is much shallower than the ice thickness of the Qiyi Glacier (tens
to hundreds of meters), which implies that their incision ability is relatively limited. Changes in the gradients of the six
supraglacial channels showed a relatively consistent pattern because they were influenced by the complex undulating
topography of the glacier surface (Fig. 6). The mean gradient of the six supraglacial channels ranges from 0.22 to 0.41, with
channel E having the steepest mean gradient. For a 10 m elevation interval, the maximum channel gradient (0.81) was observed
at an elevation of 4,640 m in channel E, while the minimum channel gradient (0.09) was found in channel F near the glacier-
covered mountain pass.

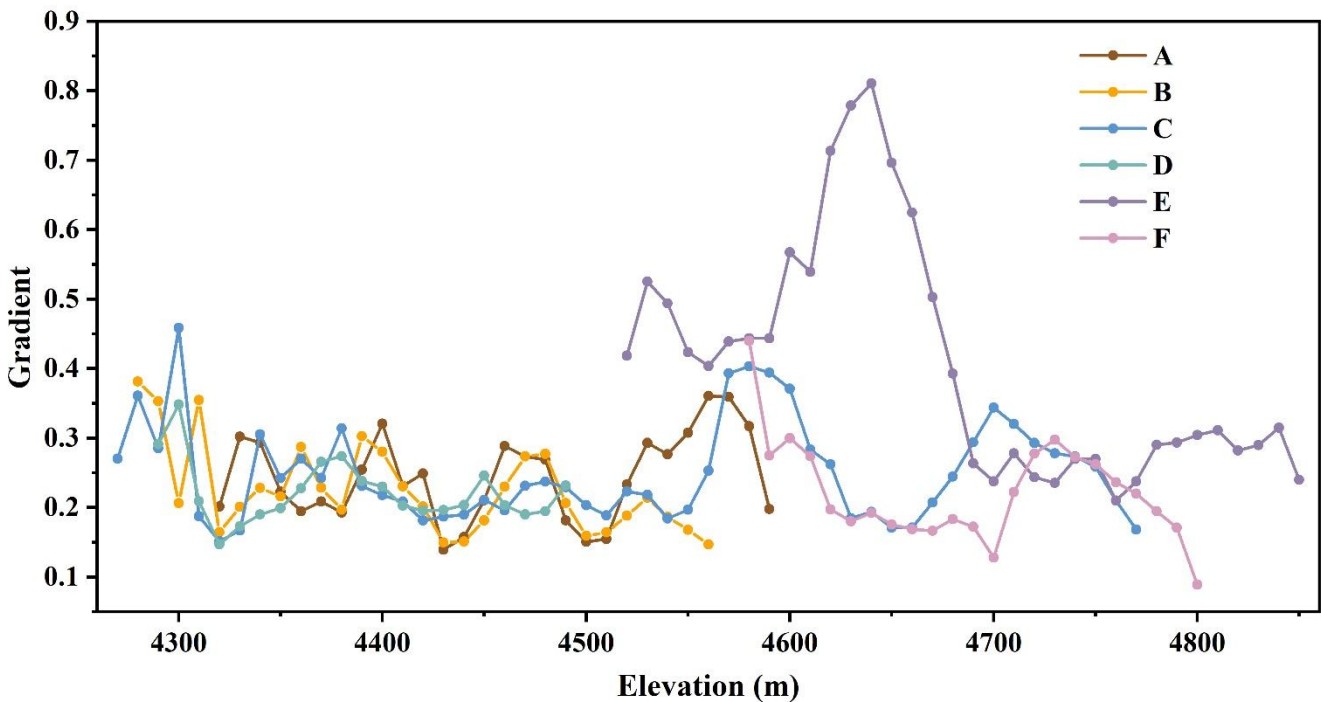

**Figure 6.** Variations in the gradient of six supraglacial channels with elevation at a 10 m interval (the labeling A to F corresponds to channels A to F as shown in Fig. 4a).

### 3.2.3 Variations in the geometry of the channel cross section

Both channel width and depth of all six supraglacial channels generally show decreasing trends with increasing elevation (Fig. 7a and 7b), with the maximum channel width and depth (1.25 and 1.00 m, respectively) observed near the channel outfall at the glacier terminus. In general, for a given channel, its width increases with depth and vice versa; however, exceptions are found for the lower reaches of channel C, which does not have the widest channel near the outlet with the deepest channel. Another noteworthy point is that, instead of the outlet, channel F has the deepest and relatively wider channel at its headwater at 4,800 m.a.s.l., possibly because its headwater is a shallow supraglacial lake over a relatively flat mountain pass that provides warmer meltwater to the channel.

Assuming a constant ratio of water depth to channel depth (water depth + $\Delta h$), the ratio of width to $\Delta h$ was calculated as a surrogate for the true channel width/depth (*W/D*) ratio. In general, the *W/D* ratio increased with elevation for all channels, except the upper reaches of channel F, which may have been influenced by the supraglacial lake at the headwater (Fig. 7c). The distribution pattern of the *W/D* ratio with elevation suggests that regions with a larger discharge in the lower part of the glacier may experience stronger downward erosion than lateral erosion.

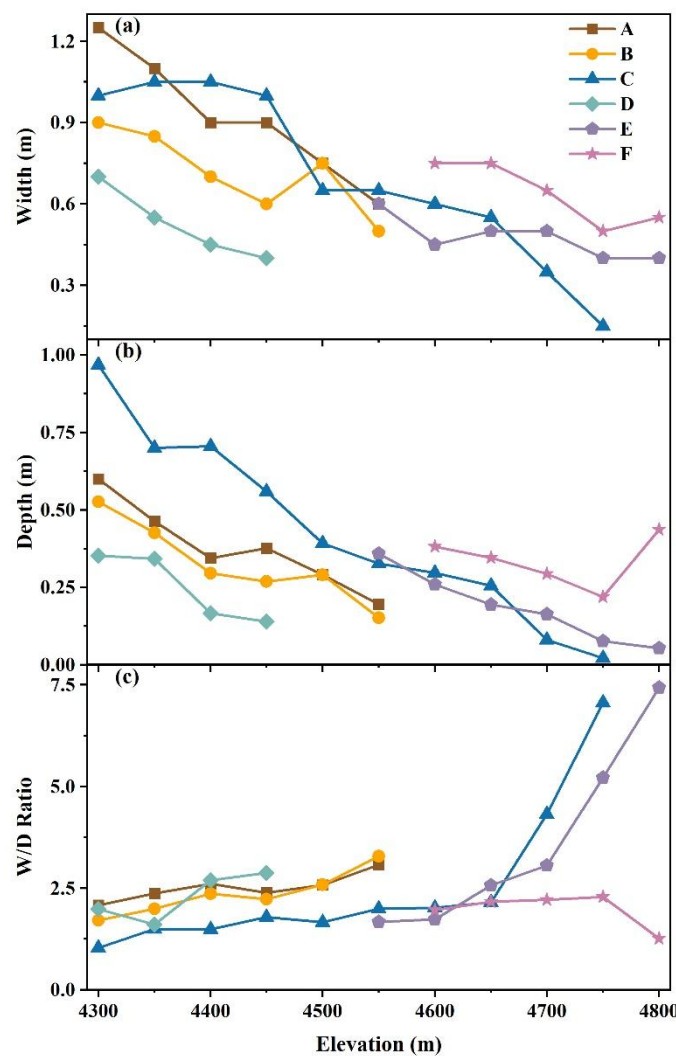

231

**Figure 7.** Variations in (a) width, (b) depth, and (c) *W/D* ratio of six supraglacial channels with elevation changes at a 50 m elevation interval.

### 3.3 Relationships between channel geometry and annual discharge

Both the Pearson and Spearman correlations showed that annual discharge ($Q_a$) was significantly positively correlated with sinuosity, lateral deviation, channel width and depth, but significantly negatively correlated with width to depth ratio(W/D) and gradient (Table 1). The water potential energy ($GQ_a$) showed a correlation pattern similar to that of discharge. Among all the geometric parameters, channel width showed the strongest correlation with discharge in both Pearson ($r = 0.82$) and Spearman ($\rho = 0.86$) correlation analyses, followed by lateral deviation ($r = 0.80$, $\rho = 0.82$), which was very similar to those of water potential energy despite of some slight differences. This suggests that both the channel profile and the meander of

supraglacial channels were closely correlated with the discharge and water potential energy, and thus could be considered as
indicators of changes in them.
Table 1. Pearson's ($r$) and Spearman's rank ($\rho$) coefficients of geometric parameters with discharge $Q_a$ and water potential energy $GQ_a$, all
coefficients that are significant at a minimum level of 0.05, 0.01, and 0.001 are labeled with *, **, and ***, respectively.

| | Sinuosity | | Lateral deviation | | Width | | Depth | | W/D | | Gradient | |
|---|---|---|---|---|---|---|---|---|---|---|---|---|
| | $r$ | $\rho$ | $r$ | $\rho$ | $r$ | $\rho$ | $r$ | $\rho$ | $r$ | $\rho$ | $r$ | $\rho$ |
| $Q_a$ | 0.73*** | 0.80*** | 0.80*** | 0.82*** | 0.82*** | 0.86*** | 0.74*** | 0.76*** | -0.44*** | -0.51*** | -0.20** | -0.21** |
| $GQ_a$ | 0.77*** | 0.82*** | 0.81*** | 0.79*** | 0.76*** | 0.82*** | 0.74*** | 0.74*** | -0.43*** | -0.53*** | 0.15* | 0.10 |


Partial correlation analysis, after excluding the effects of discharge and water potential energy, further revealed an inherent
connection between the geometric parameters (Table 2). When discharge was considered the control variable, the highest
correlation coefficient was found between sinuosity and lateral deviation (0.75), followed by that between width and depth
(0.70). When water potential energy was used as the control variable, the highest correlation was observed between width and
depth (0.72), as well as between sinuosity and lateral deviation (0.72). Regardless of whether discharge or water potential
energy was used as the controlling factor, after removing their influence, strong correlations remained between sinuosity and
lateral deviation, width and depth, indicating that both channel meander and profile characteristics exhibited stable internal
variation patterns. Furthermore, the relatively weak correlation between channel meander characteristics and width or depth
showed that their variations may not be synergistic, and thus can be jointly used to estimate discharge. These findings provided
a crucial reference for the selection of independent variables in discharge regression models.
Table 2. The partial correlation coefficient between each two geometric parameters, with discharge $Q_a$ and water potential energy $GQ_a$ as
control variables. Significance levels of 0.05, 0.01, and 0.001 are denoted by *, **, and ***, respectively.

| | Sinuosity | | Lateral deviation | | Width | | Depth | | W/D | |
|---|---|---|---|---|---|---|---|---|---|---|
| | $Q_a$ | $GQ_a$ | $Q_a$ | $GQ_a$ | $Q_a$ | $GQ_a$ | $Q_a$ | $GQ_a$ | $Q_a$ | $GQ_a$ |
| Lateral deviation | 0.75*** | 0.72*** | | | | | | | | |
| Width | 0.18* | 0.20** | 0.09 | 0.16* | | | | | | |
| Depth | 0.29*** | 0.24** | 0.26*** | 0.24*** | 0.70*** | 0.72*** | | | | |
| W/D | -0.07 | -0.05 | 0.04 | 0.05 | -0.47*** | -0.46*** | -0.53*** | -0.53*** | | |
| Gradient | 0.20** | -0.14 | 0.13 | -0.28*** | 0.13 | -0.27*** | 0.29*** | -0.08 | -0.24** | -0.07 |


## 3.4 Regression models of annual discharge of the supraglacial channels

As stated in Section 2.5, both linear and nonlinear regression models were used to fit the annual discharge $Q_a$ of the supraglacial channels using geometric parameters with significant correlations with $Q_a$. For the linear regression method, all variables were initially included to obtain an optimal fit of $Q_a$; then, a stepwise regression method was used to establish a more efficient linear fitting formula by excluding variables with minimal impact on the regression results. For the nonlinear regression method, the focus was on the robustness of the NLS model, which was fitted using a minimal number of parameters that are easier to obtain without significantly reducing the simulation results.

As illustrated in Fig. 8a, the linear regression model incorporating all variables ($SIN$, $LD$, $G$, $W$, $D$, $\beta$, and Elevation) showed a similar $R^2$ value to that of the stepwise regression model including only $LD$, $G$, and $W$ (0.780 vs. 0.782), with RMSE of 0.096 and $0.097 \times 10^6$ m$^3$, MAE of 0.077 and $0.076 \times 10^6$ m$^3$, and nRMSE of 10.64% and 10.60%, respectively. This indicates that the inclusion of Elevation, $D$, and $\beta$ in the linear regression model does not improve the explanation of the variation in the annual discharge. The stepwise regression model was expressed as follows:

$$Q_{step} = 0.230 \times LD - 0.232 \times G + 0.493 \times W - 0.181. \tag{3}$$

In contrast to the difficulty in obtaining width $W$ in Eq. (3) from high-spatial-resolution satellite images, $LD$ and $SIN$ are the most easily and accurately extracted supraglacial channel characteristics from the digitized river network. According to the correlation analysis described in Section 3.3, the $r$ values of $LD$ and $SIN$ with $GQ_a$ were both improved compared with those of $Q_a$, particularly for $LD$, which showed the highest $r$ value with $GQ_a$. Therefore, it may be more appropriate to develop a regression model using a nonlinear relationship. $LD$ was selected as the independent variable using a simple linear regression model.

$$GQ_a = a \times LD + b, \tag{4}$$

where $a$ and $b$ are the regression coefficients. The nonlinear equation can be obtained by transforming Eq. (4) as follows:

$$Q_{NLS} = a \times \frac{LD}{G} + \frac{b}{G} + c, \tag{5}$$

where the regression coefficients are $a = 0.108$, $b = -0.061$, and $c = 0.137$ in our study area and $Q_{NLS}$ is the annual discharge predicted using the nonlinear model. The $R^2$, RMSE, MAE, and nRMSE values in Eq. (5) are 0.662, $0.122 \times 10^6$ m$^3$, $0.091 \times 10^6$ m$^3$, and 12.87%, respectively.

When compared with the observed annual discharge, both the linear and nonlinear models exhibited large fluctuations. Therefore, the stepwise and NLS methods were smoothed using a five-point moving average, and the filtered results were more consistent than the original curves, with $R^2$ increasing to 0.818 and 0.814, respectively. Their RMSE, MAE, and nRMSE values were 0.091 and $0.099 \times 10^6$ m$^3$, 0.073 and $0.077 \times 10^6$ m$^3$, and 12.58% and 15.11%, respectively. The NLS model is more robust and can be more widely applied as its input parameters require only $LD$ and the channel gradient $G$, which can be easily calculated from the DEM data.

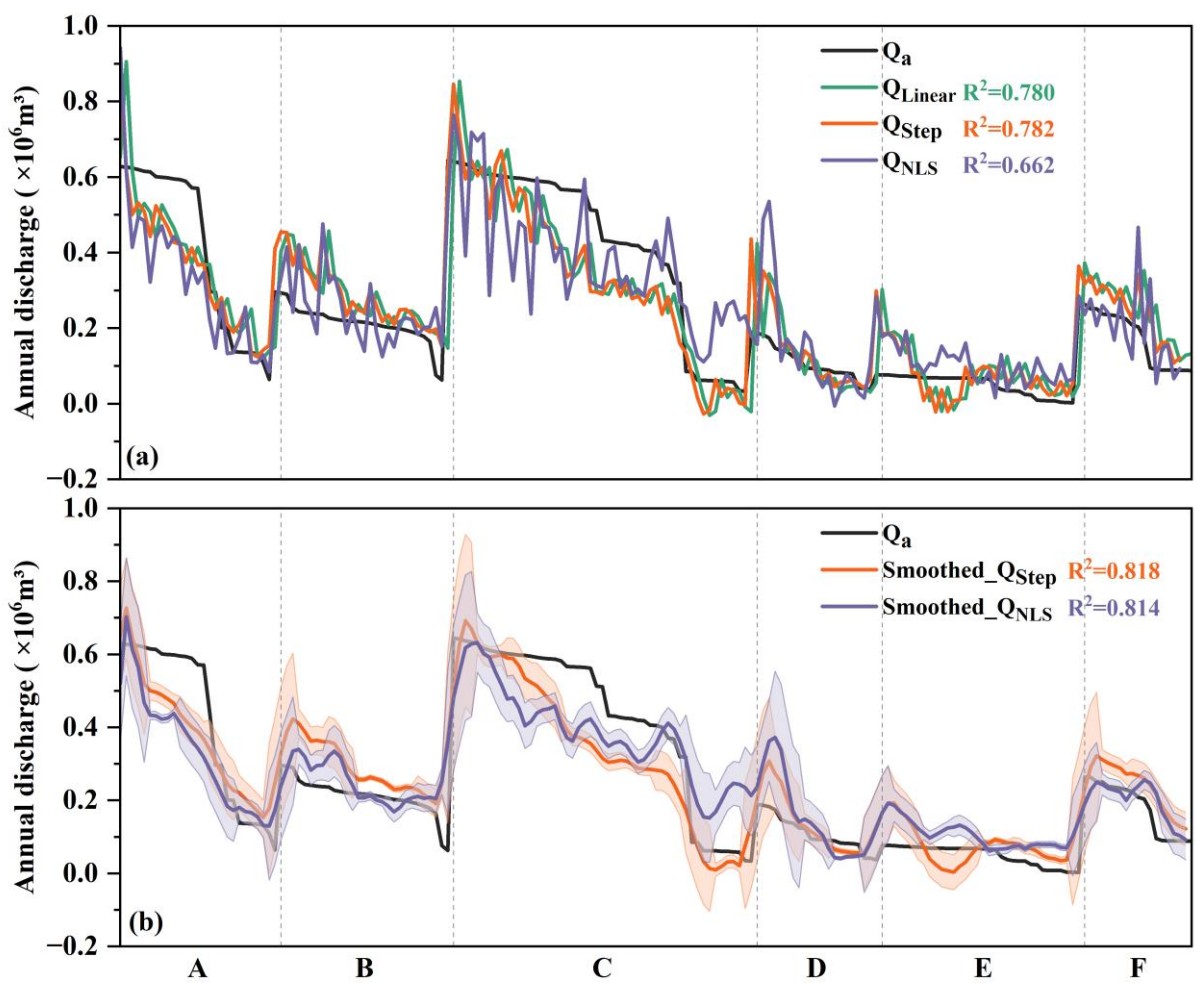

289

**Figure 8.** Comparison between observed annual discharge ($Q_a$) and (a) original results and (b) five-point moving average fitted results using linear, stepwise, and NLS regression models. The shaded error bands indicate twice the standard deviation range. The $x$ axis is arranged at an order of increasing elevation for each channel meander, and each channel is divided by vertical gray dashed lines.

## 4 Discussion

Smith et al. (2015) highlighted that the processes of meltwater generation, storage and transport on the surface of the Greenland Ice Sheet "remain one of the least studied hydrological processes on Earth". This knowledge gap is also significant for mountain glaciers. The extreme lack of field observations and challenges associated with data collection have resulted in significant uncertainties in our understanding and estimation of meltwater discharge volume from mountain glaciers, particularly in the Tibetan Plateau region. Previous studies have primarily estimated supraglacial runoff through direct discharge observations or runoff models driven by meteorological data (Muthyala et al., 2022; Yang et al., 2019). Yang and Smith (2016) demonstrated

that catchment area is a dominant control on runoff, with larger basins generally producing greater discharge. However, when catchments originate from high elevation mountain glaciers, steep and complex glacier surfaces may weaken the dominant relationship between catchment area and glacier runoff. The supraglacial channel morphology is more strongly influenced by local topography, which is one of the reasons we retained gradient as a variable. Building on established physical principles, given the significant correlations of $Q_a$ and $GQ_a$ with the geometric parameters of the supraglacial channels (Table 1, 2 and Fig. 8a, 8b), we innovatively used these geometric parameters to predict the surface meltwater discharge in mountain glacier and establish quantitative relationship between supraglacial channel geometry and annual glacier runoff. Our results are significant for estimating meltwater discharge over mountain glaciers with complex topography. In the following sections, we will further discuss the uncertainties of the supraglacial channel parameters, limitations of the regression models, and the applicability of these models.

**4.1 Uncertainties in supraglacial channel parameters**

The sinuosity and lateral deviation of supraglacial channels can be derived from DEM data based on hydrological analysis or digitization of the orthomosaic map (Fig. 2a). If the spatial resolution requirements of visible-light images are satisfied, the sinuosity and lateral deviation are the easiest parameters to obtain. However, the identification of other parameters such as river width, depth and gradient requires DEM data with a higher spatial resolution (Fig. 2b). While the width is discernible in certain regions from visible light images (e.g., the wider parts of the river at the lower reaches of the glacier), discriminating the spectral character between the narrower channel and the adjacent bank over the upper and middle parts of the glacier remains a challenge, making it difficult to extract complete width data across the entire glacier based on the visible light image alone.

In addition, during the glacial ablation season, part of the supraglacial river channel is often occupied by flowing water, making it difficult to derive the true channel depth from UAV-based DEM data. Therefore, the UAV-based depth ($\Delta h$) represents the channel depth by considering the water surface as the "bottom" of the glacier (Fig. 2b). This might exert a greater influence in the shallower channel over the upper glacier but a relatively minor influence for the deeper channel over the lower parts. This study assumes a constant ratio of true channel depth to depth above the water surface (i.e., $\Delta h$), although this assumption inevitably introduces some error when compared to actual conditions. Furthermore, the channel depth was not included in the stepwise regression model, but this does not necessarily mean that the true channel depth cannot improve the predictive ability of the regression model. Nevertheless, the depth of the supraglacial channel depends on centimeter-level DEM data, which remains a challenge given the current development trend of high-resolution commercial optical satellites.

The channel gradient must also be derived from DEM data. In regions without UAV data, deriving the channel gradient from lower resolution DEMs (e.g., 30 m GDEM, SRTM) requires careful evaluation. Therefore, sinuosity and lateral deviation are currently the most promising and accurate supraglacial channel parameters that can be obtained using commercial optical satellite imagery at centimetre scale resolution.

## 4.2 Uncertainties in the calculation of glacier annual discharge and regression model

The reliability of glacier discharge calculations depends heavily on the accuracy of the observed glacier mass balance and precipitation data. The estimation of the distributed glacier mass balance is hampered by a lack of true validation values, which complicates the error assessment of spatial patterns in mass balance calculations based on single-point measurements of ablation stakes (Kaser et al., 2006; Xu et al., 2019). Specifically, the uncertainties of stake-based mass balance can be classified into three groups: (1) errors in field observations; (2) errors related to spatial extrapolation over the entire glacier; and (3) error due to unaccounted interannual changes in glacier area (Dyurgerov et al., 2002). Among these, (1) is very small (centimeter scale) and negligible at the annual scale, and (3) typically arises in multi-year studies. Since this study focuses solely on Qiyi Glacier's 2023 data, it is also excluded. The primary error is spatial extrapolation from ablation stakes, which typically requires higher resolution DEMs (e.g., obtained by TLS laser scanning (Xue et al., 2024)) for assessment, but such data were unavailable for Qiyi Glacier in 2022–2023 due to the absence of the UAV-derived high-resolution DEM in 2022. However, given the geographic proximity, similar glacier types, and comparable elevation range covered by stakes, we adopted the error assessment from Urumqi Glacier No.1 reported by Xu et al. (2019) as a reference, where the error of traditional stake-based mass balance was about ±0.12 m w.e., within 17%.

Regarding the error in the precipitation pattern over mountain glaciers, although the elevation effect of precipitation is obtained from in situ precipitation observations, the influence of wind and local topography also introduces some uncertainty in the precipitation distribution. To evaluate the combined effect of uncertainties in mass balance and precipitation on glacier discharge calculations, the meltwater volume of each pixel of the Qiyi Glacier was manually changed by a range of ±5%, ±10%, and ±20% for both stepwise regression and NLS model results. Based on the reference from Xu et al. (2019), we set the maximum uncertainty range of our stake-based mass balance estimates at ±20%. Under the maximum uncertainty scenario (20%), when compared with the original models without additional errors, the mean absolute error (MAE) for both stepwise and NLS methods remained very small (0.092 and $0.109 \times 10^6$ m³). The RMSE increased slightly from 0.097 and $0.122 \times 10^6$ m³ to 0.117 and $0.146 \times 10^6$ m³, while the nRMSE rose from 10.60% and 12.87% to 19.68% and 17.06% (Fig. 9), showing reasonable robustness of the two current regression methods. These results indicate that although spatial extrapolation uncertainty is the dominant error source, it does not affect our finding that channel geometry can be used to estimate glacier runoff.

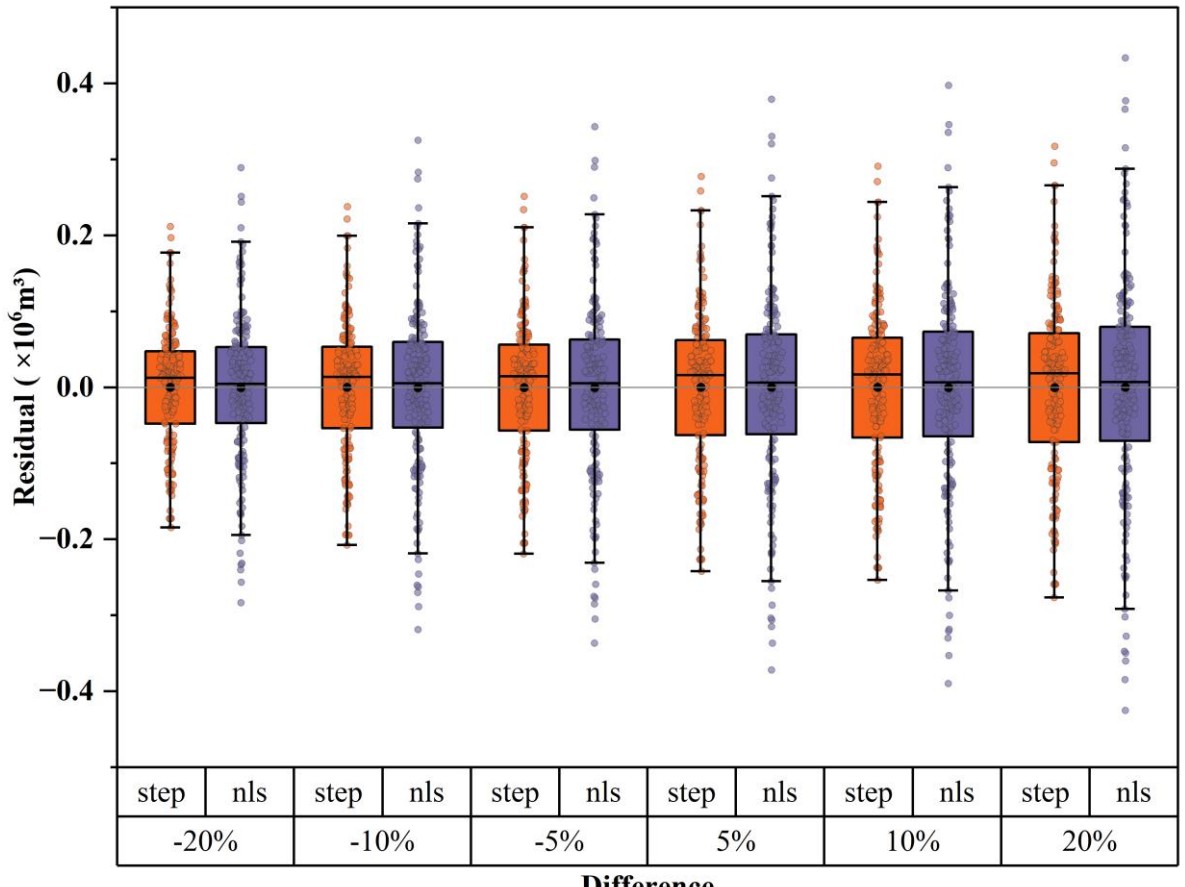

**Figure 9.** Residual distribution for two regression methods under three artificial error scenarios (±5%, ±10%, and ±20%). The black dots and black short lines of the box plots are the mean and median of the residuals, respectively; the whiskers denote 1.5 times the IQR from both ends of the box.

## 4.3 Limitations of statistical regression methods

There is still a lack of studies on the physical theories of the formation of each morphological characteristic, which hinders us from proposing an ideal nonlinear statistical regression model framework, even after several attempts. For instance, we tried to consider the effect of ice temperature on erosion intensity by incorporating normalized elevation as well as more complex regression models from the physical symbolic optimization (PhySO) framework, but their performance did not improve evidently compared to our simple regression models. According to a theoretical model of channel meander formation (Karlstrom et al., 2013), for a given supraglacial channel with a fixed cross section, runoff, and ice temperature, a bender channel meander tends to produce more heat by flowing water, which favors ice melting and lateral migration at the vertex. This implies that, in addition to the influence of external forces, such as water discharge, the channel morphology itself has a complex self-feedback process. Although Pitcher and Smith (2019) emphasized the importance of integrating thermal forcing

models (ice and water temperature), topography, and hydraulic parameters because they are helpful in understanding the
physical mechanism behind the formation of supraglacial river systems, the framework of physical-based models or more
appropriate forms of nonlinear models that can help improve the performance of predicted discharge using only the geometric
parameters of channels are still unavailable.

## 4.4 Applicability and perspective of regression model

The Qiyi Glacier is a typical subcontinental glacier on the Tibetan Plateau with weak subglacial hydrological processes. As a
result, meltwater mainly originates from the glacier surface and flows across the surface, forming a relatively continuous
channel network. Thus, we infer that this approach is well suited to polythermal or cold-based glaciers with limited subglacial
hydrological activity or surface structures. However, it may be less applicable to maritime glaciers with large crevasses or
numerous active moulins, such as those on the southern Tibetan Plateau and the Greenland Ice Sheet. But the model still has
positive implications for supraglacial river segments between crevasses and moulins. The drainage areas of supraglacial river
segments can be obtained by watershed delineation, and the meltwater discharge of each drainage area can be estimated from
their geometry.
It should also be emphasized that, while channel geometric parameters can be derived from remote sensing data alone, the
establishment of this regression model require mass balance data for calibrating the coefficients in runoff estimation due to
scaling differences among glaciers. Therefore, the broader application of this method to other glaciers must be accompanied
by reliable mass balance data.

## 5 Conclusions

Accurate estimation of glacier meltwater runoff is of great importance for quantifying the effects of climate change on glacier
dynamics and evaluating the impacts of glacier changes on downstream water resource security. Based on high-precision UAV
orthophotos and LiDAR data, this study derived a comprehensive supraglacial channel network over the Qiyi Glacier for the
first time, from which a series of channel morphological characteristics were used to predict the meltwater discharge using
different regression methods. Compared with traditional methods based on field observations or glacier runoff models, our
study provides a novel perspective for accurately estimating annual glacier discharge for larger glaciated areas without
meteorological and hydrological observations. The stepwise regression model incorporating the parameters of lateral deviation,
channel gradient, and width of the supraglacial channel performed better, explaining approximately 78.2% of the variation in
the observed meltwater discharge. The regression accuracy improved to 81.8% after the five-point moving average filtering.
The NLS regression model using only the more readily available parameters of lateral deviation and channel gradient
performed slightly less well, explaining 66.2% of the variation in meltwater discharge, whereas the explained variance was
improved to 81.4% after applying a five-point moving average. In recent years, centimeter-scale multi-satellite constellations
with predominantly onboard optical sensors, such as the WorldView Legion satellites, Pelican-2 satellite, and Gen-3
constellation, have become an important development trend. Our study indicated that although the skill of the stepwise
regression model incorporating lateral deviation, channel gradient, and width was higher than that of the NLS regression model
using only lateral deviation and channel gradient, their performances were nearly the same after five-point moving average
filtering. When using centimeter scale resolution satellite remote sensing data to predict glacier meltwater yield over larger
areas, it is recommended to use a nonlinear fitting model with a minimum amount of input data and higher reliability.

## Author Contribution

Conceptualisation: YW. Methodology: LX, YW, AC. Investigation: LX, YW, AC. Project administration: NW, SZ. Resources:
NW, SZ. Software: HS. Visualisation: LX, HS. Writing (original draft): LX. Writing (review and editing): YW.

## Competing interests

The contact author has declared that none of the authors has any competing interests.

## Data Availability Statement

The UAV-processed data (orthomosaic and DEM), supraglacial channel line vectors, and regression data (including geometric
parameters and regression results) are available online (Xie and Wu, 2025).

## Acknowledgements

This work was supported by the National Natural Science Foundation of China (Grant No. 42171139, 42130516), the Second
Tibetan Plateau Scientific Expedition and Research Program (2019QZKK020102), and the Open Research Fund of National
Cryosphere Desert Data Center (Grant No.2024NCDC001).

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
