# Peer review of "Estimation of annual runoff using supraglacial channel geometry derived from UAV surveys of Qiyi Glacier, northern Tibetan Plateau"

_EGUsphere, 2025_

## Author Response (AR1)

Dear Dr. Bagshaw,

Thank you for inviting us to submit a revised version of our manuscript. Please find below a point-by-point reply summary of the revisions we have made in response to reviewer suggestions. The reviewer comments colored black and our responses colored red.

**Reviewer 1**

Dear Dr. Storrar,
Thank you very much for the time and effort devoted to reviewing our manuscript and we are grateful for your insightful and constructive comments. Below, we provide a point-by-point reply to each comment.

This paper presents the results of a detailed UAV survey of Qiyi glacier, with the intention of taking morphometric measurements of supraglacial channels that can, in turn, be used to estimate runoff. The analysis reveals relationships between sinuosity and lateral deviation (similar metrics), gradient and discharge. This relationship is promising as a means of deriving runoff information from parameters that are measurable by high-resolution satellite remote sensing. Other parameters (height/width) also showed strong(er) relationships, though these are dependent on UAV data and so less widely applicable to satellite remote sensing methods.
This is a neatly conceptualized and well-executed study and the manuscript is well-written. The model has the potential to be very useful for estimating runoff across many glaciers, subject to some caveats. The data and methods appear sound, although some of the morphometric data requires more explanation. The discussion should explain more about exactly how applicable this method is to other glaciers, since I think this is over-stated a bit in the text. I expand on these points below, but otherwise I think this paper is a very useful contribution and I enjoyed reading it.

Reply: We really appreciate your positive evaluation of our work and are pleased that the study was found to be useful and enjoyable to read. To make the content more rigorous, the applicability of this method to other glaciers has been revised in the discussion. Details will be mentioned in the reply to the fifth paragraph of your comments.

More information is needed on how exactly some of the morphometric data were calculated. Channel height and width is not straightforward to measure due to the topographic complexity of glacier surfaces, and so it is important that a consistent method is used to represent this (which I am guessing is what was done). How were the points h1 and h2 derived? This has a very significant implication for the measurement of both height and width. Please add some explanation of this to the methods section (2.3).

Reply: Thank you for pointing this out. We have revised the description in Section 2.3 to ensure readers can follow our definitions and emphasized that this method is consistent throughout this study. Specifically, supraglacial channels are continually eroded by rapid water flow, which creates relatively distinct inflection points in the cross-sectional slope. The two points with the maximum slope are designated as $h_1$ and $h_2$. A representative cross-section (channel A at 4350 m) has been included, showing the positions of $h_1$ and $h_2$ (Fig. S1). The following clarification has been added

to line 123:

*"To ensure consistency in defining channel geometry, we applied a standardized approach to determine channel width and depth. Since supraglacial channels are continuously eroded by rapid water flow, there are relatively distinct inflection points in the slope of the cross-section. The two points on the cross-section with the steepest slope gradients were designated as $h_1$ and $h_2$. Channel depth ($\Delta h$) was defined as the average vertical distance from these two points to the lowest point of the channel, whereas channel width (W) was defined as the horizontal distance between $h_1$ and $h_2$. A representative cross-section (channel A at 4350 m) has been included in the Supplement (Fig. S1) to illustrate the positions of $h_1$ and $h_2$."*

[Figure]

**Figure S1.** Cross-section of River A at 4350m elevation, the red dashed line represents the profile's slope.

Since the premise of this study is that supraglacial channel morphometry may be a better way of estimating runoff than existing modelling techniques, it would be good to see some comparison of the data generated here with modelled runoff data. This is hinted at in the conclusions (line 365). I don't know how feasible this is, but if possible would be a useful addition.

Reply: Thank you for your suggestions. First, runoff estimated from in situ ablation stakes and precipitation measurements is commonly used as a verification for glacier runoff models. For this reason, we did not perform a comparison with other modelled runoff results. Second, while modeling methods have the advantage of being applicable to other glaciers, our study area is limited to Qiyi Glacier. Although we currently have in situ ablation stake and precipitation data, other meteorological data required to drive runoff models are lacking; therefore, we did not conduct additional runoff model simulations in the present work. In our future research, we plan to install additional meteorological instruments, which will allow us to compare field measurements, runoff

model simulations, and channel geometry methods in a systematic way. Thank you again for your valuable feedback.

The discussion of applicability (section 4.4) should be extended to discuss further exactly how typical this glacier is, and other types of glacier to which this model probably does not apply besides tidewater glaciers. For example, temperate glaciers, glaciers with strong surface structures (e.g. crevasses) and debris-covered glaciers, all of which will be very common, are likely too complex to be represented by this model. This section should also note that to derive similar relationships for other glaciers requires mass balance data at the very least (because the scaling relationships are likely different for each glacier), so unless I am mistaken it can't be applied using remote sensing alone.

Reply: We fully agree with your suggestions. Following your advice, we have reorganized the discussion in Section 4.4 to further emphasize the representativeness of the Qiyi Glacier, and we have expanded our description of glacier types where this model may be difficult to apply or may require substantial adjustments, including temperate glaciers, debris-covered glaciers, and glaciers with strong surface structures. The following revisions have been made in the Line 392:
*"However, the applicability of this approach is limited for some other glacier types. For example, on maritime glaciers with extensive crevasses and relatively stronger subglacial hydrological processes (e.g., in the southern Tibetan Plateau and the Greenland Ice Sheet), some supraglacial rivers often terminate in crevasses or moulins, which makes our regression model less suitable for estimating glacier runoff. Similarly, temperate glaciers, debris-covered glaciers, and glaciers with strong surface structures are likely too complex to be represented without substantial modifications to the model."*
  In addition, although channel geometric parameters can be derived from remote sensing alone, the establishment of this method still requires mass balance data to calibrate the coefficients for runoff estimation, due to scaling differences among glaciers. We have therefore explicitly stated this at line 401:
*"It should also be emphasized that, while channel geometric parameters can be derived from remote sensing data alone, the establishment of this regression model require mass balance data for calibrating the coefficients in runoff estimation due to scaling differences among glaciers. Therefore, the broader application of this method to other glaciers must be accompanied by reliable mass balance data."*

I suggest rewording the title to make it clearer to something like: "Estimation of annual runoff using supraglacial channel geometry derived from UAV surveys of Qiyi Glacier, northern Tibetan Plateau"

Reply: The authors all agree that your suggested title more precisely and fully reflects the research content of the article. We have revised the title accordingly.

Line 14-15: Re-order words (novel remote sensing method)

Reply: Change to *"a novel remote sensing method"*.

Line 15: Not sure what is meant by 'discharge volume'? I think you just mean discharge here?

Reply: We acknowledge that this was an incorrect statement, and the term "volume" has been removed.

Lines 24-25: I don't understand the point here. We don't have centimetre-resolution satellite imagery?

Reply: We sincerely apologize for the lack of clarity in our previous wording. What we intended to convey is that if the remote sensing imagery and DEM for a specific glacier study area reach meter-level resolution, our method can serve as an effective solution for analyzing glacial runoff changes. The original sentence has been revised to:
*"If satellite remote sensing data with meter-level spatial resolution are available for a specific glacier research area, our regression models, based solely on the UAV-derived supraglacial channel network, will be a promising solution for monitoring changes in annual glacier discharge."*

Line 59: Suggest 'uncrewed' instead of 'unmanned' to remove gender bias.

Reply: We have made the suggested changes.

Lines 68-71: Are these findings published? If so, please provide a reference.

Reply: The reason we did not include citations to our field investigation results is that these findings are currently based on observational experience and data that have not yet been formally published. The research in this paper also confirms that our field observations are reliable.

Line 72: Discharge?

Reply: Change "discharge volume" to "annual discharge".

Line 129: I assume from the equation that mass balance is expressed with negative values indicating mass *loss* specifically? It would be good to clarify this in the text below the equation (as well as stating the units).

Reply: Thank you for pointing this out. We have added clarification under Equation 1:
*"mass balance is expressed in millimeters water equivalent (mm w.e.), with negative values indicating mass loss."*

Line 155: Pre-existing ice structure (e.g. fractures) also exerts a strong influence on channel morphology (e.g. Rippin et al. 2015: https://onlinelibrary.wiley.com/doi/full/10.1002/esp.3719?casa_token=5KxM3AvAxIYAAAAA%3AAODpVkdle62Mntm4D44VdOLPP_C7as8R1utImLL7u3vRhY6XGGyBfY0zhIDLpP5UlRx0olELoZHKRCU)

Reply: This is a very valid point. We have integrated this concept into Section 2.5(line 179)

Line 185: Median and *mean*?

Reply: The term "mean" is used to denote the average value. We have replaced "averages" in the text (line 209 and 214).

Line 292: 'especially for mountain glaciers' needs to be in a separate sentence because it is not what Smith said.

Reply: We have restructured the sentence to separate the phrase "especially for mountain glaciers" into a new sentence.

Line 296-7: Change glaciers to glacier (it has only been done at one!).

Reply: We have made the suggested changes.

Line 308-9: Yes, sinuosity and lateral deviation could be determined from high-resolution satellite imagery, but you also need gradient, which can be taken from DEMs, which will be at significantly lower resolution where no UAV data are available. That is not to say that it is not useable, but perhaps worth stating.

Reply: Your suggestions have helped us make the manuscript more rigorous. We have clarified that gradient must be derived from DEM data, and that in regions without UAV data, gradient extracted from lower-resolution DEMs (e.g., 30 m GDEM, SRTM) still requires further evaluation regarding their applicability. Nevertheless, such datasets can provide valuable scientific reference for future research. We have added a justification to line 339, as follows:
*"The channel gradient must also be derived from DEM data. In regions without UAV data, channel gradient derived from lower-resolution DEMs (e.g., 30 m GDEM, SRTM) still require further evaluation regarding their applicability. Therefore, sinuosity and lateral deviation are currently the most promising and accurate supraglacial channel parameters that can be obtained using commercial optical satellite imagery at a centimeter resolution."*

Reviewer 2

Dear Reviewer,
We sincerely thank you for your comprehensive and detailed review of our manuscript. Your comments have been invaluable in helping us to refine our work, we will carefully address the issues you raised.

My main concern with this manuscript is that the authors claim they develop a "novel method for estimating annual discharge" of supraglacial channels which is not supported by work presented in the manuscript. Specifically,

1.The authors calculation of annual glacial discharge (which per the title is the pivotal contribution of this work) relies on data from two weather stations and nine ablation stakes. These 9 ablation stakes are used to calculate mass balance (constraining MB in equ 1) at a 5 cm resolution. There is a brief discussion of some errors in the discussion but they cite errors from other studies and not an error analysis for this work. Moreover, the manuscript lacks details on the methodology for ablation stake measurements (what dates and at what frequency were ablation stakes measured?, what is the associated error?) And the data itself is not presented in the manuscript.

Reply: Thank you for your insightful review and constructive feedback. For glacier runoff:

1. Glacier runoff models based on coarse-resolution meteorological data are commonly used, but the uncertainty remains significant. Generally, the algebraic sum of mass balance derived from ablation stakes and precipitation provides more reliable results. We have now added a detailed description of the ablation stake measurements to line 153, Section 2.4 (including measurement dates and frequency):

*"The stakes were measured at annual intervals. Several new ablation stakes were installed on July 20, 2022, and subsequently measured and maintained on August 24, 2023. Detailed information for each stake data, including stake ID, position, elevation, and mass balance records, are provided in Supplement (Table S2). The uncertainty of the mass balance based on the ablation stakes is discussed in detail in Section 4.2."*

2. The detailed data have been added to Supplement (Table S2), including stake ID, elevation, and mass balance records.

3. As you point out, the uncertainties of stake-based mass balance can be classified into three groups: (1) errors in field observations; (2) errors related to spatial extrapolation over the entire glacier; and (3) error due to unaccounted interannual changes in glacier area (Dyurgerov et al., 2002). Among these, class (1) is very small (centimeter scale) and negligible at the annual scale, and (3) typically arises only in multi-year studies. Since this study focuses solely on Qiyi Glacier's 2023 data, it is also excluded. Therefore, the primary error is spatial extrapolation from ablation stakes, which typically requires higher resolution DEMs (e.g., obtained by TLS laser scanning (Xue et al., 2024)) for assessment. Due to the absence of the UAV-derived high resolution DEMs for Qiyi Glacier during 2022 and 2023, we cannot provide a site-specific error estimate. However, given the geographic proximity, similar glacier types, and comparable elevation range covered by stakes, we used the error assessment from Urumqi Glacier No.1 reported by Xu et al. (2019) (±0.12 m w.e.) as a reference. Calculations showed their interpolation errors within 17%, so we set the maximum uncertainty range at ±20% in our discussion (Section 4.2) based on this result. From these results

we conclude that, although spatial extrapolation uncertainties are the dominant error source, they do not affect our finding that channel geometry can be used to estimate glacier runoff. The uncertainty analysis added to line 347 is as follows:

*"Specifically, the uncertainties of stake-based mass balance can be classified into three groups: (1) errors in field observations; (2) errors related to spatial extrapolation over the entire glacier; and (3) error due to unaccounted interannual changes in glacier area (Dyurgerov et al., 2002). Among these, (1) is very small (centimeter scale) and negligible at the annual scale, and (3) typically arises in multi-year studies. Since this study focuses solely on Qiyi Glacier's 2023 data, it is also excluded. The primary error is spatial extrapolation from ablation stakes, which typically requires higher resolution DEMs (e.g., obtained by TLS laser scanning (Xue et al., 2024)) for assessment, but such data were unavailable for Qiyi Glacier in 2022–2023 due to the absence of the UAV-derived high-resolution DEM in 2022. However, given the geographic proximity, similar glacier types, and comparable elevation range covered by stakes, we adopted the error assessment from Urumqi Glacier No.1 reported by Xu et al. (2019) as a reference, where the error of traditional stake-based mass balance was about ±0.12 m w.e., within 17%."*

**Table S2.** Ablation stake measurements record

| Stake ID | Latitude (°) | Longitude (°) | Elevation (m) | MB (mm w.e.) |
|---|---|---|---|---|
| 3-1 | 39.249607 | 97.753264 | 4320 | -1894.8 |
| 4-2 | 39.247748 | 97.753548 | 4350 | -1540.2 |
| 5-1 | 39.246667 | 97.754925 | 4380 | -1280.7 |
| 6-5 | 39.244167 | 97.753611 | 4430 | -935.7 |
| 7-1 | 39.243494 | 97.755908 | 4480 | -1236.9 |
| 8-1 | 39.241767 | 97.758044 | 4550 | -1249.5 |
| 9-1 | 39.240086 | 97.759672 | 4610 | -945.6 |
| 10-4 | 39.237303 | 97.764169 | 4700 | -919.2 |
| 12-mountain pass | 39.234425 | 97.768039 | 4800 | -594.3 |

2.Estimating annual discharge into a supraglacial stream is not a novel method, and even if it were, it would need to be validated on actual discharge measurements. Intuitively, streams with larger catchment areas should convey a greater proportion of annual discharge (e.g., Yang and Smith 2016), however, this isn't specifically described and therefore makes it seem like that is a novel finding of this work. I recommend revising the manuscript to take care to properly cite and describe known physical relationships and by potentially adding a section to the discussion which distinguishes between new findings (stream morphology relationships with discharge) vs. findings that align with previous studies.

Reply: As you noted, there are indeed existing studies that have calculated supraglacial stream runoff, but these are mostly based on observed discharge or runoff models driven by meteorological data (Muthyala et al., 2022; Yang et al., 2019). Our intention here is to highlight the novelty of using supraglacial channel geometric parameters (e.g., sinuosity and lateral deviation) to estimate runoff, rather than relying solely on hydrological models or in situ discharge measurements.

In addition, we have expanded the discussion to emphasize the topographic differences between the Greenland Ice Sheet (studied by Yang and Smith (2016)) and high elevation mountain glaciers.

When catchments originate from high elevation mountain glaciers, steep and complex glacier surfaces may weaken the simple dominant relationship between catchment area and glacier runoff, and supraglacial channel geometry is more strongly influenced by local topography. This is one of the reasons why we keep the gradient variable in our regression equations. Finally, we have carefully implemented your suggestions to explicitly acknowledge the established physical principles you mentioned (contribution of catchment areas to glacier runoff) and to clearly distinguish between previously known findings and the novel results of our study in the Discussion, line 308:

*"Previous studies have primarily estimated supraglacial runoff through direct discharge observations or runoff models driven by meteorological data (Muthyala et al., 2022; Yang et al., 2019). Yang and Smith (2016) demonstrated that catchment area is a dominant control on runoff, with larger basins generally producing greater discharge. However, when catchments originate from high elevation mountain glaciers, steep and complex glacier surfaces may weaken the dominant relationship between catchment area and glacier runoff. The supraglacial channel morphology is more strongly influenced by local topography, which is one of the reasons we retained gradient as a variable. Building on established physical principles, we focus on supraglacial channel geometric parameters (e.g., sinuosity and lateral deviation) as predictors of glacier runoff, and establish quantitative relationship between supraglacial channel geometry and annual glacier runoff."*

The main contribution of this work seems to be the geometric analysis performed on the very high resolution DEM created for Qiyi Glacier in 2023. By framing the analysis in the context of annual discharge, which has significant yet undiscussed uncertainties, undercuts confidence in the results. I suggest revisiting the annual discharge calculation, including a more robust error analysis and discussion (in addition to increasing the detail in the methods section), and shifting the main findings of the manuscript to what can be confidently argued by your results.

Reply: Thank you for your positive feedback. This study is the first in the Tibetan Plateau to conduct high resolution geometric analysis of supraglacial rivers. We have supplemented the methodology and discussion for annual runoff calculations and error analysis in our response to your first question (Section 4.2 and Table S2). We hope these revisions meet your satisfaction.

Methodology on the Automatic determination of supraglacial streams and "manual correction" should be elaborated on. How many streams were originally identified by the algorithm? How many needed to be corrected visually? Only 11 streams are shown in Fig 4a so it is unclear why an automatic method needed to be employed in the first place.

Reply: The reasons we first conducted an automatic extraction method are as follows: the automatic determination algorithm ensures an objective and repeatable identification of the entire supraglacial stream network. Regarding the number of streams: if each tributary and main stem is counted as one stream from its headwater, the DEM-based automated extraction algorithm identifies not only the 11 clearly visible streams shown in Fig. 4a, but also an additional 22 lower-order, smaller, and shorter tributaries that are much less distinct in the figure. As for manual correction, for these smaller tributaries, although stream lines can be delineated based on DEM data, they are not clearly visible when overlaid on orthophotos (with a resolution of 5 cm).Therefore, under the principle that

"streams automatically extracted from DEM must be clearly distinguishable on orthophotos to ensure the realism of stream morphology," the purpose of manual correction was to truncate these smaller tributaries and upstream sections where channels are difficult to recognize. We sincerely thank you again for your detailed inquiry. We have expanded the description of the channel automatic extraction at line 134:

*"The automated algorithm identified a total of 33 channel segments, including not only the six main channels (labeled A–F in Fig. 4a) but also smaller, low-order tributaries. To ensure the realism of the extracted channels, all automatically derived streams were carefully compared with the orthophotos, channels that could not be clearly identified in the orthophotos were truncated, particularly small tributaries and river sections at the upstream headwaters."*

Section 3.4 belongs in the methods section

Reply: Thank you for your suggestion. After careful consideration, we have decided to retain this section in the Results. In Section 2.5 we briefly introduce the regression model, although the regression model and related contents in Section 3.4 have methodological aspects, the specific coefficients, equations, and errors are one of the core findings of our research. We therefore believe presenting them here may offer readers a more coherent and fluid narrative. To ensure clarity, we have added explicit cross references between Sections 2.5 and 3.4, allowing readers to easily locate the complete model details.

L54: Yes, not every study uses weather station data (as cited in this sentence) but many studies do use weather station data, this sentence is quite misleading in this regard.

Reply: We thank the reviewers for pointing out this overgeneralization. The sentence was intended to emphasize the challenge of lacking field meteorological data in remote areas, but the phrasing was not sufficiently precise. The statement has now been revised:

*"Estimates of meltwater runoff at basin or larger scales have often relied either on the limited field observations (Gleason et al., 2016; Smith et al., 2017, 2021) or on glacier runoff models driven by coarse-resolution climate data (Beamer et al., 2016; Hock, 2005; Sicart et al., 2008; Wang et al., 2024; Yang et al., 2025). Although in situ meteorological observations from weather stations are used in some glacier studies, such data are often unavailable or sparse for remote glacierized regions, leading to uncertainties in modeled runoff that hinder a robust quantitative analysis of its relationship with supraglacial channel geometry."*

Tables, what do the ** mean? I can't find a description for this in the text.

Reply: we sincerely apologize for missing that detail, In the table, the asterisks denote the significance levels of the correlation coefficients (* = $p < 0.05$, ** = $p < 0.01$, *** = $p < 0.001$). We have added an explanation of this in the captions of Tables 1 and 2.

Fig 9. Move to methods

Reply: We appreciate your suggestion. After discussion, we believe it is more appropriate to retain

Figure 9 in its current location (Section 3.4). This figure provides an intuitive illustration of the regression model performance and its comparison with the observed values, making it more suitable as a result presentation in Section 3.4.

**References:**

Beamer, J. P., Hill, D. F., Arendt, A., and Liston, G. E.: High-resolution modeling of coastal freshwater discharge and glacier mass balance in the Gulf of Alaska watershed, Water Resour. Res., 52, 3888–3909, https://doi.org/10.1002/2015WR018457, 2016.

Dyurgerov, M., Meier, M., and Armstrong, R. L.: Glacier mass balance and regime: Data of measurements and analysis, Institute of Arctic and Alpine Research, University of Colorado, Boulder, CO, USA, 2002.

Gleason, C. J., Smith, L. C., Chu, V. W., Legleiter, C. J., Pitcher, L. H., Overstreet, B. T., Rennermalm, A. K., Forster, R. R., and Yang, K.: Characterizing supraglacial meltwater channel hydraulics on the Greenland Ice Sheet from *in situ* observations, Earth Surf. Process. Landf., 41, 2111–2122, https://doi.org/10.1002/esp.3977, 2016.

Hock, R.: Glacier melt: a review of processes and their modelling, Prog. Phys. Geogr., 29, 362–391, 2005.

Muthyala, R., Rennermalm, Å. K., Leidman, S. Z., Cooper, M. G., Cooley, S. W., Smith, L. C., and van As, D.: Supraglacial streamflow and meteorological drivers from southwest Greenland, The Cryosphere, 16, 2245–2263, https://doi.org/10.5194/tc-16-2245-2022, 2022.

Sicart, J. E., Hock, R., and Six, D.: Glacier melt, air temperature, and energy balance in different climates: The Bolivian Tropics, the French Alps, and northern Sweden, J. Geophys. Res. Atmospheres, 113, 2008JD010406, https://doi.org/10.1029/2008JD010406, 2008.

Smith, L. C., Yang, K., Pitcher, L. H., Overstreet, B. T., Chu, V. W., Rennermalm, Å. K., Ryan, J. C., Cooper, M. G., Gleason, C. J., Tedesco, M., Jeyaratnam, J., van As, D., van den Broeke, M. R., van de Berg, W. J., Noël, B., Langen, P. L., Cullather, R. I., Zhao, B., Willis, M. J., Hubbard, A., Box, J. E., Jenner, B. A., and Behar, A. E.: Direct measurements of meltwater runoff on the Greenland ice sheet surface, Proc. Natl. Acad. Sci., 114, E10622–E10631, https://doi.org/10.1073/pnas.1707743114, 2017.

Smith, L. C., Andrews, L. C., Pitcher, L. H., Overstreet, B. T., Rennermalm, Å. K., Cooper, M. G., Cooley, S. W., Ryan, J. C., Miège, C., Kershner, C., and Simpson, C. E.: Supraglacial River Forcing of Subglacial Water Storage and Diurnal Ice Sheet Motion, Geophys. Res. Lett., 48, e2020GL091418, https://doi.org/10.1029/2020GL091418, 2021.

Wang, L., Liu, H., Bhlon, R., Chen, D., Long, J., and Sherpa, T. C.: Modeling glacio-hydrological processes in the Himalayas: A review and future perspectives, Geogr. Sustain., 5, 179–192, https://doi.org/10.1016/j.geosus.2024.01.001, 2024.

Xu, C., Li, Z., Li, H., Wang, F., and Zhou, P.: Long-range terrestrial laser scanning measurements of annual and intra-annual mass balances for Urumqi Glacier No. 1, eastern Tien Shan, China, The Cryosphere, 13, 2361–2383, https://doi.org/10.5194/tc-13-2361-2019, 2019.

Xue, Y., Hong, X., He, X., Kang, S., Wang, S., and Ding, Y.: An evaluation of TLS-based glacier change assessment in the central Tibetan plateau, Int. J. Digit. Earth, 17, 2375527, https://doi.org/10.1080/17538947.2024.2375527, 2024.

Yang, K. and Smith, L. C.: Internally drained catchments dominate supraglacial hydrology of the

southwest Greenland Ice Sheet, J. Geophys. Res. Earth Surf., 121, 1891–1910, https://doi.org/10.1002/2016JF003927, 2016.

Yang, K., Smith, L. C., Fettweis, X., Gleason, C. J., Lu, Y., and Li, M.: Surface meltwater runoff on the Greenland ice sheet estimated from remotely sensed supraglacial lake infilling rate, Remote Sens. Environ., 234, 111459, https://doi.org/10.1016/j.rse.2019.111459, 2019.

Yang, Z., Bai, P., Tian, Y., and Liu, X.: Glacier Coverage Dominates the Response of Runoff and Its Components to Climate Change in the Tianshan Mountains, Water Resour. Res., 61, e2024WR037947, https://doi.org/10.1029/2024WR037947, 2025.

---

## Editor Decision (ED1)

Dear authors,

Thank you for your thorough response to reviews and revision of the manuscript. I find that you have addressed Reviewer 2's main concerns through the following statement '*Previous studies have primarily estimated supraglacial runoff through direct discharge observations or runoff models driven by meteorological data (Muthyala et al., 2022; Yang et al., 2019). Yang and Smith (2016) demonstrated that catchment area is a dominant control on runoff, with larger basins generally producing greater discharge. However, when catchments originate from high elevation mountain glaciers, steep and complex glacier surfaces may weaken the dominant relationship between catchment area and glacier runoff. The supraglacial channel morphology is more strongly influenced by local topography, which is one of the reasons we retained gradient as a variable. Building on established physical principles, we focus on supraglacial channel geometric parameters (e.g., sinuosity and lateral deviation) as predictors of glacier runoff, and establish quantitative relationship between supraglacial channel geometry and annual glacier runoff'* which highlights the novelty of your method, and the applicability to high mountain glaciers.

However, there are a few minor amendments that would improve understanding of your error calculations, and also enable readers to understand the future applicability of the method. Please find my suggestions below.

I look forward to reading the next version of the manuscript.

Dr Liz Bagshaw, Editor.

L93: 'According to our regular meteorological, hydrological, and mass balance observations (June to August) starting in 2002, the Qiyi Glacier has suffered from an obviously increasing ice-melting trend since 2016 (Chen et al., 2024)'.

Rephrase: 'we have undertaken regular meteorological, hydrological, and mass balance observations on Qiyi Glacier from June to August since 2002, which have demonstrated an increase in melt, particularly since 2016 (Chen et al. 2024).'

I have some concerns about your response to the reviewer on their query of mean and median. This may be a translation issue, but mathematically, mean and median are both averages. I would like assurance that your statement *'Reply: The term "mean" is used to denote the average value. We have replaced "averages" in the text (line 209 and 214).'* Is correct, and that you do report mean.

L317: 'of great importance' is a very subjective statement without evidence. Please rephrase to 'significant', 'useful' or something more circumspect.

L340: grammatical improvement: 'In regions without UAV data, deriving the channel gradient derived from lower resolution DEMs (e.g., 30 m GDEM, SRTM) requires careful evaluation'

L342: 'commercial optical satellite imagery at a centimeter resolution' – are there commercial satellites with a single centimetre resolution? Instead suggest 'at centimetre scale resolution'

L367: 'These results indicate that, although spatial extrapolation uncertainty isthe dominant error source, it does not affect our finding that channel geometry can be used to estimate glacier runoff.'

I find this statement a bit vague. Instead, could you give an idea of the practical change in discharge estimates with your maximum error? I know this is in the boxplots, but it would be useful to see an example value in the text.

L392 'However, the applicability of this approach is limited for some other glacier types. For example, on maritime glaciers with extensive crevasses and relatively stronger subglacial hydrological processes (e.g., in the southern Tibetan Plateau and the Greenland Ice Sheet), some supraglacial rivers often terminate in crevasses or moulins, which makes our regression model less suitable for estimating glacier runoff. Similarly, temperate glaciers, debris-covered glaciers, and glaciers with strong surface structures are likely too complex to be represented without substantial modifications to the model'

To me, this discounts quite a significant proportion of glaciers! Could you rephrase this paragraph to highlight the glaciers it can do, rather than those it can't? For example: 'this approach is very well suited to polythermal or cold-based glaciers with limited subglacial hydrological activity or surface structures. However, it is likely to struggle in regions with large crevasses or numbers of active moulins, such as....'

L414: 'five-point moving filtering' should there be a 'mean' or 'average' in this sentence?

---

## Author Response (AR2)

Dear Dr. Liz Bagshaw,

Thank you very much for your positive feedback and thoughtful suggestions on our revised manuscript. We greatly appreciate your recognition of our work's novelty and applicability to high mountain glaciers. We have carefully addressed all your suggestions and revised the manuscript accordingly. Please find our detailed responses below.

We believe that these revisions have improved the clarity and scientific quality of our manuscript. Thank you again for your time and valuable guidance that helped us further refine our paper. We look forward to your further consideration.

Sincerely,

Longjiang Xie

(On behalf of all co-authors)

L93: 'According to our regular meteorological, hydrological, and mass balance observations (June to August) starting in 2002, the Qiyi Glacier has suffered from an obviously increasing ice melting trend since 2016 (Chen et al., 2024)'.

Rephrase: 'we have undertaken regular meteorological, hydrological, and mass balance observations on Qiyi Glacier from June to August since 2002, which have demonstrated an increase in melt, particularly since 2016 (Chen et al. 2024).'

Response: Thank you for the suggestion. We have rephrased the sentence as:

"we have undertaken regular meteorological, hydrological, and mass balance observations on Qiyi Glacier from June to August since 2002, which have demonstrated an increase in melt, particularly since 2016 (Chen et al., 2024)."

I have some concerns about your response to the reviewer on their query of mean and median. This may be a translation issue, but mathematically, mean and median are both averages. I would like assurance that your statement 'Reply: The term "mean" is used to denote the average value. We have replaced "averages" in the text (line 209 and 214).' Is correct, and that you do report mean.

Response: Thank you for your careful attention. We sincerely apologize for the confusion caused by our earlier wording. We confirm that in our study, all reported "mean" values indeed refer to the arithmetic mean. Both mean and median are forms of averages. The term "averages" was used in the initial draft due to translation inconsistency. We have carefully checked the entire manuscript to ensure accuracy and clarity.

L317: 'of great importance' is a very subjective statement without evidence. Please rephrase to 'significant', 'useful' or something more circumspect.

Response: Thank you for pointing this out. As recommended, we have revised this phrase to "significant" to maintain objectivity.

L340: grammatical improvement: 'In regions without UAV data, deriving the channel gradient derived from lower resolution DEMs (e.g., 30 m GDEM, SRTM) requires careful evaluation'

Response: We have made the suggested changes.

L342: 'commercial optical satellite imagery at a centimeter resolution' – are there commercial satellites with a single centimetre resolution? Instead suggest 'at centimetre scale resolution'

Response: Thank you for your suggestion. We have revised the sentence as: "can be obtained using commercial optical satellite imagery at centimetre scale resolution."

L367: 'These results indicate that, although spatial extrapolation uncertainty is the dominant error source, it does not affect our finding that channel geometry can be used to estimate glacier runoff.' I find this statement a bit vague. Instead, could you give an idea of the practical change in discharge estimates with your maximum error? I know this is in the boxplots, but it would be useful to see an example value in the text.

Response: We appreciate this helpful suggestion. We have calculated the specific error value of runoff estimation under maximum error conditions to illustrate the practical influence of maximum uncertainty:
"Under the maximum uncertainty scenario (20%), when compared with the original models without additional errors, the mean absolute error (MAE) for both stepwise and NLS methods remained very small (0.092 and $0.109 \times 10^6$ m³). The RMSE increased slightly from 0.097 and $0.122 \times 10^6$ m³ to 0.117 and $0.146 \times 10^6$ m³, while the nRMSE rose from 10.60% and 12.87% to 19.68% and 17.06% (Fig. 9), showing reasonable robustness of the two current regression methods. These results indicate that although spatial extrapolation uncertainty is the dominant error source, it does not affect our finding that channel geometry can be used to estimate glacier runoff."

L392 'However, the applicability of this approach is limited for some other glacier types. For example, on maritime glaciers with extensive crevasses and relatively stronger subglacial hydrological processes (e.g., in the southern Tibetan Plateau and the Greenland Ice Sheet), some supraglacial rivers often terminate in crevasses or moulins, which makes our regression model less suitable for estimating glacier runoff. Similarly, temperate glaciers, debris-covered glaciers, and glaciers with strong surface structures are likely too complex to be represented without substantial modifications to the model'
To me, this discounts quite a significant proportion of glaciers! Could you rephrase this paragraph to highlight the glaciers it can do, rather than those it can't? For example: 'this approach is very well suited to polythermal or cold-based glaciers with limited subglacial hydrological activity or surface structures. However, it is likely to struggle in regions with large crevasses or numbers of active moulins, such as….'

Response: We fully agree with your suggestions. Following your advice, we have rewritten the paragraph to emphasize the glacier types where our approach performs well:
"Thus, we infer that this approach is well suited to polythermal or cold-based glaciers with limited subglacial hydrological activity or surface structures. However, it may be less applicable to maritime glaciers with large crevasses or numerous active moulins, such as those on the southern Tibetan

Plateau and the Greenland Ice Sheet."

L414: 'five-point moving filtering' should there be a 'mean' or 'average' in this sentence?

Response: You're absolutely correct. We have added "average" to clarify that the method used is "five-point moving average filtering".

**Technical corrections in File validation**

1) Table and figure numbering of the supplement are independent from each other. Hence, please renumber the single table to "Table S1" (instead of "Table S2")

Response: We have renumbered the supplementary table as Table S1.

2) Please ensure that the colour schemes used in your maps and charts allow readers with colour vision deficiencies to correctly interpret your findings. Please check your figures using the Coblis – Color Blindness Simulator (https://www.color-blindness.com/coblis-color-blindness-simulator/) and revise the colour schemes accordingly. --> Figs. 5, 6, 7

Response: The colour schemes in Figs. 4–7 have also been revised after checking with the Coblis – Color Blindness Simulator to ensure accessibility for readers with colour vision deficiencies.